

# Methodology for constructing a flood-hazard map for a future climate

Yuki Kimura[1][2] , Yukiko Hirabayashi[3], Yuki Kita[2][4], Xudong Zhou[2], Dai Yamazaki[2]

[1]Risk Assessment Department, MS&AD InterRisk Research & Consulting, Inc., 2-105, Kanda Awajicho, Chiyoda-ku,
Tokyo 101-0063, Japan
[2]Institute of Industrial Science, The University of Tokyo, 4-6-1, Komaba, Meguro-ku, Tokyo 153-8505, Japan
[3]Department of Civil Engineering, Shibaura Institute of Technology, 3-7-5 Toyosu, Koto-ku, Tokyo, 135-8548, Japan
[4]Gaia Vision Inc. 6-23-4 Jingumae, Shibuya-ku, Tokyo, 150-0001, Japan

*Correspondence to*: Yuki Kimura (yuuki.kimura@ms-ad-hd.com)

**Abstract.** Flooding is a major natural hazard in many parts of the world, and its frequency and magnitude are projected to increase with global warming. With increased concern over ongoing climate change, more detailed and precise information about climate-change risks is required for formulating local-scale countermeasures. However, the impacts of biases in climate-model outputs on river-flood simulation have not been fully evaluated, and thus evaluation of future flood risks using hazard maps (high-resolution spatial distribution maps of inundation depths) has not been achieved. Therefore, this study examined methods for constructing future-flood-hazard maps and discussed their validity. Specifically, we compared the runoff-correction method that corrects for bias in general-circulation-model (GCM) runoff using the monthly climatology of reanalysis runoff with the lookup method, which uses the GCM simulation results without bias correction to calculate changes in the return period, and depends on the reanalysis simulation to determine absolute flood depths. The results imply that the runoff-correction method may produce significantly different hazard maps compared to those based on reanalysis of runoff data. We found that in some cases, bias correction did not perform as expected for extreme values associated with the hazard map, even under the historical climate, as the bias of extreme values differed from that of the mean value. we found that the direction of the change in future hazard (increase or decrease) obtained using the runoff-correction method relative to the reference reanalysis-based hazard map may be inconsistent with the changes projected by the GCMs in some cases. On the other hand, we confirmed that the lookup method can produce future-hazard maps that are consistent with the changes in flood risk projected by the GCMs, indicating the possibility of obtaining reasonable inundation-area distribution. The results imply that the lookup method is more suitable for future-flood hazard-map construction than the runoff-correction method. The lookup method also has the advantage of facilitating research on efficient construction of future-climate hazard maps, as it allows for improvement of the reanalysis hazard map through upgrading of the model and separate estimation of changes due to climate change. We discuss future changes in inundation areas and the affected population within the inundation area. Using the lookup method, the total population living in modeled inundation areas with flood magnitudes exceeding the 100-year return period under a future climate would be around 1.8 billion. In the assessment of future climate risks, we found that an affected population of around 0.5 billion may be missed if the historical hazard map is



used as an alternative to constructing future-hazard maps and only frequency changes are considered. These results imply that in global flood-risk studies, future-hazard maps are important for proper estimation of climate-change risks, rather than

assessing solely changes in the frequency of occurrence of a given flood intensity.

## 1 Introduction

Flooding is a common major natural hazard in many parts of the world, and its frequency and magnitude are projected to increase with global warming. The 6th Assessment Report of the Intergovernmental Panel on Climate Change (IPCC) Chapters 11 indicated that the incidence of heavy rainfall has increased in many regions since 1950 (Seneviratne et al., 2021).

Hirabayashi et al., 2021 assessed changes in the frequency of flood risk in the future and showed that flood risk will increase in many regions, including South Asia, Southeast Asia, Northeast Eurasia, eastern and low-latitude Africa, and South America. According to Dottori et al., 2018, with a temperature increase of 1.5°C, human losses from flooding could rise by 70–83% and direct flood damage by 160–240% in the absence of future adaptation measures. Alfieri et al., 2017 showed that with 4°C warming, countries representing more than 70% of the global population and global gross domestic product (GDP)

would face increases in flood risk of more than 500%. While these global-scale studies have projected total future flood losses at the continental, regional or national scales, they do not provide local flood risk information under future climatic conditions.

Following the increase in concern about ongoing climate change, detailed and high-resolution information about climate-change risk that can be used for local-scale countermeasures is essential. To elucidate the potential impacts of flood disasters,

a high-resolution map of potential disaster impacts must be developed, commonly named a hazard map. The Sendai Framework for Disaster Risk Reduction 2015–2030 produced by the United Nations Office for Disaster Risk Reduction (UNISDR) also highlighted the importance of developing hazard maps to clarify disaster risk. Flood-hazard maps describe the spatial distribution of potential inundation depths for an event of specific occurrence probability to illustrate quantitatively the risk distribution at a local scale.

Local-scale flood-hazard maps have been developed by many research institutes and local governments using historical observation data and flood-inundation models, and these maps are used for various purposes. Governments and municipalities use flood-hazard maps to identify risks and formulate business-continuity plans (BCPs) (De Moel et al., 2009). In many countries, flood-hazard maps are made available to the public to assist residents in identifying their own risks and facilitate evacuation activities. The private sector is also making progress in using local-scale flood-hazard maps. For

example, the insurance industry uses local-scale hazard maps to set premium rates corresponding to local hazards. The National Flood Insurance Program (NFIP) in the United States uses flood-insurance-rate maps (flood surface elevations for the N-year return period) provided by the Federal Emergency Management Agency (FEMA) to calculate flood-insurance-premium rates (FEMA, 2018). Companies also use local-scale hazard maps to identify flood risks to their own buildings and factories, scrutinize the contents of their insurance policies, and formulate BCPs in preparation for a possible disaster (Japan



Institute of Country-ology and Engineering 39th report, 2021). However, in many developing countries in Asia and Africa, detailed local hazard maps are unavailable.

In the research field, large-domain flood-hazard maps have been developed to assess flood risks and their distribution at the global scale. Examples of large-domain flood-hazard maps include maps constructed by Fathom (Sampson et al., 2015), the Joint Research Centre (JRC, 2022), Global Assessment Report (GAR, 2015), and Aqueduct Floods published by the

World Research Institute (WRI) (Aqueduct, 2022). Uses of large-domain flood-hazard maps include estimation of the affected population within an inundation area and determination of the impacts of flooding on GDP and urban areas in the current climate (Ward et al., 2020). Validation of large-domain flood-hazard maps is currently underway. Hirabayashi et al., 2022 and Trigg et al., 2016 compared multiple global flood models and analyzed the factors contributing to differences in inundation areas and depths. Although their accuracy may be insufficient, large-domain hazard maps for the current climate

are now being used for various decision-making purposes. Hirabayashi et al., 2022 provided recommendations for the practical application of large-domain hazard maps in corporate practice.

In addition to flood-hazard maps of the historical period, developing flood-hazard maps for a future period is also essential to assess climate-change risk quantitatively. While assessment of climate-change risks (e.g. extreme temperatures, droughts and heavy-rainfall events) has been widely performed using direct output variables of general circulation models (GCMs),

such as precipitation and temperature (Li et al., 2021, Lu et al., 2019), assessment of future flood risks based on the spatial distribution of inundation depths has not yet been established. Given that inundation depths are regulated by local topography at scales much finer than the resolution of GCMs, additional processing of GCM outputs using a flood-risk model is essential. At present, no global high-resolution future-climate inundation-depth profile has been sufficiently verified.

Simulation of inundation areas and depths under a future climate at high resolution is technically possible using the latest

global river models together with climate-projection data and downscaling techniques. However, the reliability of future flood-hazard mapping has not been thoroughly assessed, in part because no methodology to correct the bias present in the runoff output from GCMs has yet been established. Climate-projection data contain biases, and the direct use of climate-projection data faces problems, including the inability to estimate the duration of inundation, which is important for estimating indirect damage (Taguchi et al., 2022). While bias-correction methods for precipitation and temperature have

been studied in detail (Watanabe et al., 2012, Hempel et al., 2013 and Lafond et al., 2014), such methods have not been established for runoff data for use as inputs to flood models.

At present, methods for constructing future flood-hazard maps have not been evaluated in detail. Therefore, this study investigated the following points to validate global flood-hazard maps under a future climate constructed using the global river model. We compared two representative methods for generating future flood-hazard maps that handle bias in GCM

runoff, and investigated which method produces the most reasonable inundation-depth distribution. The causes of differences in the hazard maps constructed using the two methods were investigated, and the most appropriate method for creating future-hazard maps was assessed. In addition, we examined how much the estimated changes in future flood risk differed depending on whether bias correction was implemented and the method used for future-hazard mapping.





This paper is organized as follows. Section 2 describes the models and bias-correction methods used in this study and the
method used to construct future flood-hazard maps. Section 3 explores several methods to construct hazard maps for a future
climate, and the cause of differences among hazard maps constructed using various methods. In Section 4, we discuss which
method is most appropriate for creating future-hazard maps and how much the estimates of future flood risk changes differed
depending on whether bias correction was implemented and on the method used for future-hazard mapping.

## 2 Methods

### 2.1 Model and Data

We used the Catchment-based Macro-scale Floodplain Model (CaMa-Flood; Yamazaki et al., 2011) ver. 4.01 (Yamazaki et
al., 2021) for global river and inundation simulations. The major advantage of CaMa-Flood is its high computational
efficiency. In CaMa-Flood, river channel flow and floodplain inundation can be calculated simultaneously using sub-grid
topographic parameters. In this model, river flow simulation is conducted on the basis of the unit catchment element, and
water level and flood extent are diagnosed from the water volume in each unit catchment using sub-grid topographic
parameters. The local inertial equation (Yamazaki et al., 2013) is used as the basic flow equation. This equation can
represent the backwater effect, which is important for accurate reproduction of inundated areas. A flow scheme for
bifurcated channels is included in CaMa-Flood (Yamazaki et al., 2014). Although CaMa-Flood is a global model, it is unique
in that it represents the physical processes necessary to reproduce floodplain inundation dynamics.

In this study, we simulated global river and inundation dynamics using three types of runoff data to construct flood-hazard
maps. The three types of runoff data were: reanalysis-based runoff data obtained from past observations (Reanalysis_Runoff),
GCM-output runoff without bias correction (GCM_Runoff_Ori), and GCM-output runoff with bias correction (GCM
_Runoff_BC). The GCM-output runoff data were analyzed for two periods: historical (1980–2014) and future (2066–2100).
The reanalysis runoff data used in this study were VIC-Bias-Corrected (Yang et al., 2021), which are the output of the
Variable Infiltration Capacity (VIC) Land Surface Model (Liang et al., 1994, Liang et al., 1996) with bias correction using
the Global Streamflow Characteristics Dataset (Beck et al., 2015) as reference data. Yang et al., 2021 compared discharge
data based on VIC-Bias-Corrected runoff with observations and found that the VIC-Bias-Corrected model has excellent
ability to estimate discharge and high reproducibility of extreme values. For VIC-Bias-Corrected runoff data, the original
spatial resolution (3-arcmin) was input to CaMa-Flood. For GCM-output runoff (GCM_ Runoff_Ori and GCM_Runoff_BC),
we used nine GCMs: MIROC6, IPSL-CM6A-LR, GFDL-CM4, NorESM2-MM, ACCESS-CM2, INM-CM5-0, MPI-ESM1-
2-HR, MRI-ESM2-0, and EC-Earth3, similar to Hirabayashi et al., 2021. GCM-output runoff was converted from its original
spatial resolution to 30-arcmin resolution through bi-linear interpolation. We also generated bias-corrected runoff, and the
methods used for bias correction are described in detail in Section 2.3. The runoff products, resolutions, and periods assessed
in this study are listed in Table 1.




**Table 1: Runoff products used in this study**

| | Reanalysis Data (Reanalysis_Runoff) | GCM without Bias Correction (GCM_Original_Runoff) | | GCM with Bias Correction (GCM_BiasCorrect_Runoff) | |
|---|---|---|---|---|---|
| **Runoff Products** | VIC-Bias-Corrected | 9GCM of CMIP6 | | 9GCM of CMIP6 | |
| **Time Period (Year)** | 1980~2014 | 1980~2014 | 2066~2100 | 1980~2014 | 2066~2100 |
| **Runoff Resolution** | 3arcmin | 30arcmin | | 30arcmin | |
| **Climate Scenarios** | Historical | Historical | Future (ssp585) | Historical | Future (ssp585) |

## 2.2 Reference Historical Flood-hazard map Generation

Flood-hazard maps were generated through downscaling of river water levels simulated by CaMa-Flood (6-arcmin resolution) to the resolution of elevation data (3 arcsec). The general procedures used for simulation and data processing are summarized in Figure 1 (a). First, we conducted a long-term historical river hydrodynamics simulation with a daily time step using observation-based runoff data (Reanalysis_Runoff) as an input to CaMa-Flood. The CaMa-Flood model produces outputs as daily time series, including river discharge, river water level, and flood extent. In this study, the annual maximum

river water level was calculated from daily river water level data and used for extreme-value analysis in the following step.

As the second step, river water levels at 6-arcmin resolution corresponding to the targeted return period were calculated. We fitted the Gumbel distribution (Zhou et al., 2021) to the time series of annual maximum river water levels using the L-moments method (Hosking, 2015). Due to the relatively small dataset, we used the Gumbel distribution, which provides more robust and stable results from small datasets than other distributions (Hirabayashi et al., 2021, Dankers, 2008). Then,

the river water levels corresponding to the targeted return period (RP) (e.g., 100-year) for each grid point were calculated from the Gumbel distribution.

We applied a simple post-processing method to the river water levels estimated in the previous step to obtain a more reasonable spatial distribution of water levels. Through fitting of individual Gumbel distributions, the upstream water surface elevation can become lower than the downstream level, causing an unrealistic reverse water slope. To avoid this issue, if a

reverse water slope was obtained in the water-level distribution, we elevated the water levels of upstream grids to match those of downstream grids. This reverse-slope revision during the hazard-mapping process is a new method proposed in this paper. If reverse-slope revision is not conducted, the backwater effect is not considered, despite the occurrence of a reverse slope, and the inundation-depth distribution may not be physically reasonable. For this reason, a novel reverse-slope revision method was applied in this study.




The third step is projecting the water levels of the target RP (e.g., 100-year) onto a high-resolution elevation map. Downscaling was performed under the assumption that the water surface elevation is uniform within each 6-min unit catchment, and thus the floodplain water depth is calculated when the terrain elevation of a 3-arcsec pixel is lower than the water surface elevation. The river network map of CaMa-Flood (6-arcmin resolution in this study) was constructed through upscaling of a high-resolution river topographic map (MERIT Hydro; Yamazaki et al., 2019). Through this procedure,

correspondence between 3-sec resolution pixels and 6-min unit catchments was preserved, allowing the water level simulated at 6-min resolution to be downscaled to match topographic data with 3-sec resolution (for details of the upscaling method, see Yamazaki et al., 2011).

**Figure 1: Simplified schematic diagrams of (a) flood-hazard map generation, (b) correction of bias of GCM runoff, and (c) RP value estimation via the lookup method.**



### 2.3 Future flood-depth estimation

In this study, two methods of estimating future flood depth were compared. The two methods are summarized in Sections
2.3.1 and 2.3.2.

### 2.3.1 Runoff-correction method

The runoff-correction method used bias-corrected GCM runoff for future simulation and calculated future flood depths corresponding to specific return periods through direct application of extreme-value analysis to the simulated flood depth time series. We investigated whether realistic future-hazard maps could be generated through bias correction of input runoff
data for CaMa-Flood. The procedure followed after CaMa-Flood simulation is the same as the reanalysis-based method described in Section 2.2. We generated historical hazard maps using bias-corrected runoff from the historical period to validate the reliability of the runoff-correction method.

Bias correction was applied using the trend-preserving method (Hempel et al., 2013) with runoff from the reanalysis data. Figure 1(b) shows a simplified schematic diagram of the bias-correction process for input runoff data. Specifically, the long-
term averages (1980–2014) of monthly runoff from the reanalysis data and GCM were calculated (Equation (1)), and a constant offset C, equal to the difference from the long-term average, was set for each month.

$$C_j = (\sum_{i=starting\ year}^{end\ year} R_{ij}^{Reanalysis} - \sum_{i=1980}^{2014} R_{ij}^{GCM})/(length\ of\ years) \qquad (1)$$

where $R_{ij}^{Reanalysis}$ indicates Reanalysis_Runoff and $R_{ij}^{GCM}$ indicates GCM_Runoff_Ori (i and j denote the year and month, respectively). This value was used for full time-series analysis of the GCM under a future climate (Equation (2)).

$$\widehat{R_{ij}^{GCM}} = C_j + R_{ij}^{GCM} \qquad (2)$$

where $R_{ij}^{GCM}$ and $\widehat{R_{ij}^{GCM}}$ are GCM_Runoff_Ori and GCM_Runoff_BC, respectively.

The trend-preserving method has been used in many studies, including The Inter-Sectoral Impact Model Intercomparison Project (ISI-MIP) (Warszawski et al., 2014), which provides a framework for comparing climate impact projections among sectors and scales. Willner et al., 2018 used the same method to estimate future climate-related economic damage and
showed that over the next 20 years, economic damage will increase by 17% globally, with China having the largest increase (82%).





### 2.3.2 Lookup Method

In the lookup method, we first calculate the change in flood probability between the historical and future periods using the original GCM runoff simulation (i.e., we estimate historical RP equivalent to the magnitude of the targeted future RP). Then,

the flood depth of the future target RP is estimated based on the flood depth of a historical flood of equivalent magnitude from the reanalysis runoff simulation. Because we use a lookup table to describe the relationship between flood depth and RP in the reanalysis simulation, this method is designated the lookup method. The aim of the lookup method is to use the GCM-based simulation results to calculate the relative change in RP, relying on the higher accuracy of the reanalysis simulation for absolute flood depth. The calculation process is illustrated in Figure 1(c) and the detailed procedures are as

follows. (1) The extreme-value analysis described above is applied to the river water levels obtained using Reanalysis_Runoff and a lookup table is calculated (river water levels corresponding to RP of 2 to 1000 years) for each grid. (2) Extreme-value analysis is also applied to the river water levels obtained with GCM_Runoff_Ori under the historical climate and a lookup table is calculated for each grid. (3) Next, the target RP river water levels under the future climate are calculated by applying extreme-value analysis to the river water levels calculated from GCM_Runoff_Ori under the future

climate. Then, with reference to the lookup table of river water levels created in step (2), the RP under the historical climate corresponding to the target RP water levels for the future climate is calculated. (4) Then, for each grid, the water level corresponding to the RP determined in (3) is obtained from the reanalysis lookup table of water levels created in step (1). (5) If a reverse slope is present in the water-level distribution created in (4), the reverse slope is revised, downscaling is conducted, and the result is used as the target RP hazard map for the future climate.

As noted above, the lookup method uses only the statistical frequency of flooding calculated using the GCMs and the value calculated from the reanalysis data is used to determine inundation depths corresponding to that frequency. As the lookup method does not use GCM_Runoff_BC and thus avoids the uncertainties associated with bias correction, including questionable results of bias correction for extreme events (Alfieri et al., 2017; Huang et al., 2014), several previous studies have employed this technique (Hirabayashi et al., 2013; Hirabayashi et al., 2021).

**3 Results**

The results obtained using the runoff-correction and lookup methods were compared from the following two perspectives. First, differences between the two methods were analyzed using a single GCM for runoff at the global scale, and the mechanisms underlying the observed differences were examined. Second, we compared the results obtained using multiple GCMs for runoff and assessed the uncertainty and robustness of the two methods.

**3.1 Comparison of methods for generating future-hazard maps**

We created a global reference historical hazard map based on reanalysis runoff data and a future-hazard map using the runoff-correction and lookup methods. Figure 2 shows the flood depths at 6-arcmin resolution prior to downscaling for the





purpose of visualization. Here, we present the results of the 100-year RP hazard map from IPSL-CM6A-LR, which is the
Coupled Model Intercomparison Project Phase 6 (CMIP6, Eyring et al., 2016) GCM that showed the maximum bias relative
to reanalysis data among nine GCMs tested. Figures 2(b)–(c) show that the direction of change from the reference historical
hazard map was the same with both methods for many rivers. Notable differences were found in northern parts of the Sahara
Desert, parts of North America (around the Mississippi and Missouri rivers) and around the Amazon River, where the two
methods showed opposite change trends. Both the runoff-correction and lookup methods used the reanalysis data as
reference data to handle errors in GCM-output runoff. Therefore, the change trends from the reference historical hazard map
were expected to be in the same direction. The future-hazard maps produced through the two methods showed opposite
trends in changes compared to the reference hazard map, implying that one of the methods may be unable to account
properly for changes in future flood risks when reanalysis data are used as the historical reference dataset.

For detailed analysis, we created hazard maps that focused on several river basins and examined the validity of the two
methods for constructing future-hazard maps. Prior to validating the future-hazard maps, we confirmed the validity of the
historical hazard map through the runoff-correction method. The following two river basins were selected for detailed
discussion in this paper: the Mekong River basin (specifically, the Chi-Mun River, a tributary of the Mekong River, Figure
3), where the two methods showed the same trend under a future climate, and the Amazon River basin (upstream of Manaus,
Figure 4), where the two methods showed opposite trends. In the runoff-correction process, the climatology of monthly
average GCM runoff for each grid was corrected toward the climatology of the monthly average reanalysis runoff; this step
reduced the absolute runoff errors in the GCM. Therefore, the historical hazard map constructed using the runoff-correction
method was expected to be similar to the reference historical hazard map. As shown in Figure 3(a)–(c), the Chi-Mun River
historical hazard map produced using the runoff-correction method was most similar to the reference hazard map. However,
underestimation for the Amazon River relative to the reference hazard map remained after bias correction (Figure 4(a)–(c)).
Bias-corrected reanalysis runoff using monthly climatology produced a significant difference in the hazard map compared to
the map based on reanalysis runoff.







Figure 2: 100-year RP inundation depth distributions for (a) global reference historical hazard map, (b) future hazard changes obtained using the runoff-correction method (using IPSL-CM6A-LR) relative to (a), and (c) future hazard changes obtained with the lookup method (using IPSL-CM6A-LR) relative to (a).





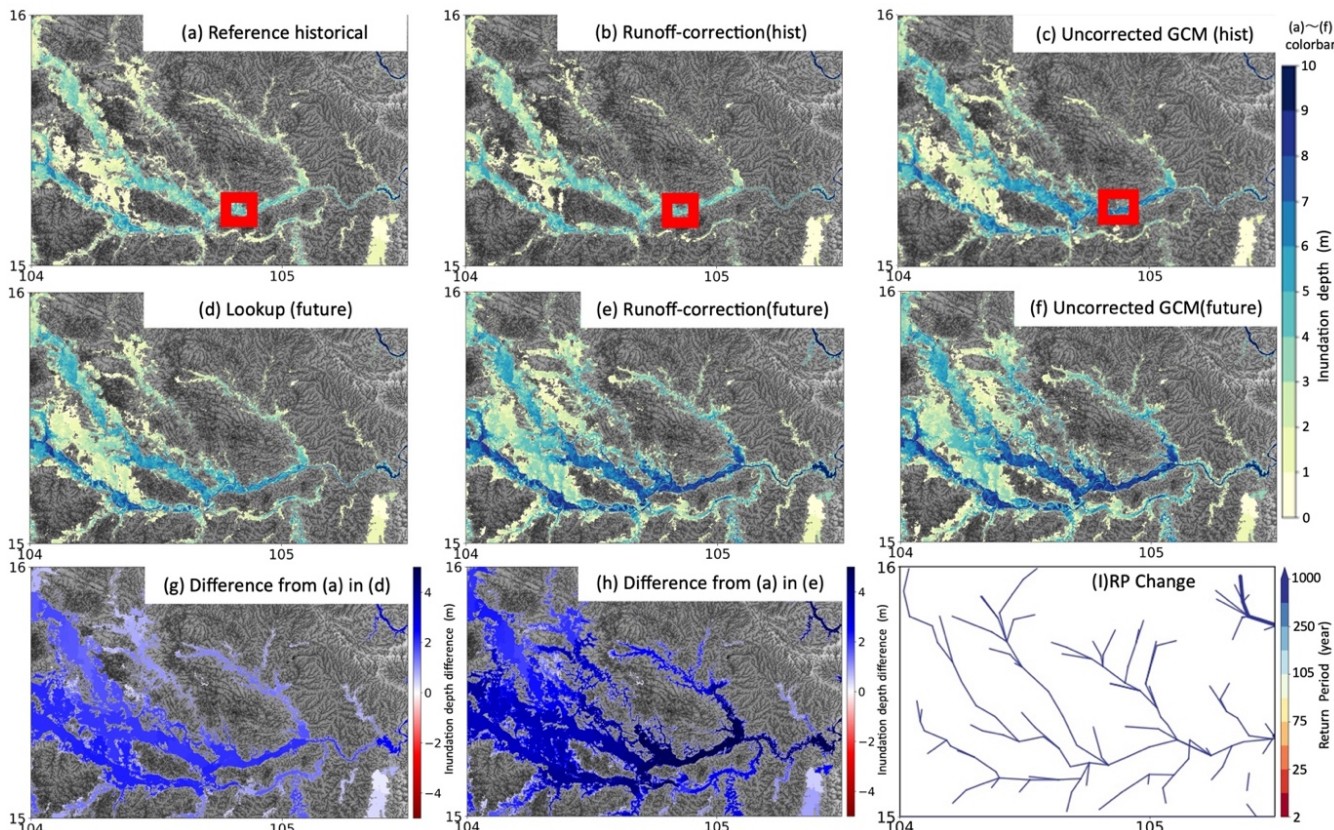

**Figure 3: 100-year RP hazard map for the Chi-Mun River basin (using the IPSL-CM6A-LR CMIP6 GCM).**

**(a) Reference historical hazard map (based on reanalysis data), (b) hazard map constructed using the runoff-correction method (historical), (c) hazard map based on uncorrected GCM runoff (historical), (d) hazard map constructed using the lookup method, (e) hazard map constructed using the runoff-correction method (future), and (f) hazard map based on uncorrected GCM runoff (future). (g–h) Differences from the reference historical hazard map in (g) the hazard map obtained with the lookup method and (h) the hazard map obtained using the runoff-correction method (future), and (i) historical return period (RP) of river water level corresponding to the 100-year RP in the future.  Red boxes in (a)–(c) indicate the location of the GRDC Ubon station (104.8617°E, 15.2217°N)**





**Figure 4: 100-year RP hazard map in Amazon River basin (using IPSL-CM6A-LR of CMIP6 GCM)**

**(a) Reference historical hazard map (based on reanalysis data), (b) hazard map constructed using the runoff-correction method (historical), (c) hazard map based on uncorrected GCM runoff (historical), (d) hazard map constructed using the lookup method, (e) hazard map constructed using the runoff-correction method (future), and (f) hazard map based on uncorrected GCM runoff (future). (g–h) Differences from the reference historical hazard map in (g) the hazard map obtained with the lookup method and (h) the hazard map obtained using the runoff-correction method (future), and (i) historical return period (RP) of river water level corresponding to the 100-year RP in the future. Black boxes in (g)–(h) indicate the location of the GRDC Itapeua station. (63.0278°W, 4.0578°S).**

The results for a future climate were assessed; as the historical climate was not corrected as expected, the future climate might have followed the same trend and failed to meet expectations. For both the Chi-Mun and Amazon rivers, using the GCM without bias correction, the shift from the historical (Figure 3(c), 4(c)) to future climate (Figure 3(f), 4(f)) implied that flood risk would increase in the future. For the Chi-Mun River, as shown in Figure 3(g) and 3(h), the future-hazard maps constructed using the runoff-correction method and lookup method showed increases in the inundation areas and depths





compared to the reference historical hazard map (Figure 3(a)). In contrast, for the Amazon River, Figure 4(h) shows that the inundation depths on the hazard map obtained using the runoff-correction method were less than the depths on the reference historical hazard map, implying that the runoff-correction method may be unable to account for the increased risk of future flooding predicted by the GCM. On the other hand, Figure 4(g) shows that inundation depths on the hazard map based on the

lookup method were greater than those on the reference historical hazard map, implying that the lookup method was able to produce hazard maps that were consistent with the changes in flood risk under a future climate projected by the GCMs. In the subsequent sections, we explore why bias correction was less effective than expected for the Amazon River.

We investigated the reasons for the differences between the reference historical hazard map and the historical hazard map constructed using the runoff-correction method. Figures 5 and 6 show the monthly mean discharge climatology, exceedance

probability curve and Gumbel distributions for the annual maximum river water levels based on CaMa-Flood simulation results using each runoff type as input values. For reference, we show the cumulative distribution function for annual maximum river water levels based on the lookup method in Figure 5(c)–(d) and Figure 6 (c)–(d). The comparison sites were Global Runoff Data Centre (GRDC) observation sites, specifically Ubon station (104.8617°E, 15.2217°N) in the Mekong River basin and Itapeua station (63.0278°W, 4.0578°S) in the Amazon River basin. The climatology of monthly average

GCM runoff for each grid was corrected toward the climatology of the reanalysis dataset. Therefore, monthly average discharge data were expected to be similar to the reanalysis data. In addition, the annual maximum river water levels were corrected, which was necessary for application of extreme-value analysis of the annual maximum river water levels when constructing the hazard maps. For these reasons, we drew an exceedance probability curve and the Gumbel distribution for the annual maximum river water levels and checked whether the values were corrected to the same scale as the reanalysis

data (Figure 5b). Prior to analysis of the Amazon River, where bias correction did not perform as expected, the Chi-Mun River was assessed. As shown in Figure 5(a)–(d), the uncorrected GCM (historical data; green dotted line) showed different behavior from the reanalysis data (black line), but the bias-corrected GCM (green line) data were similar to the reanalysis data, as expected.





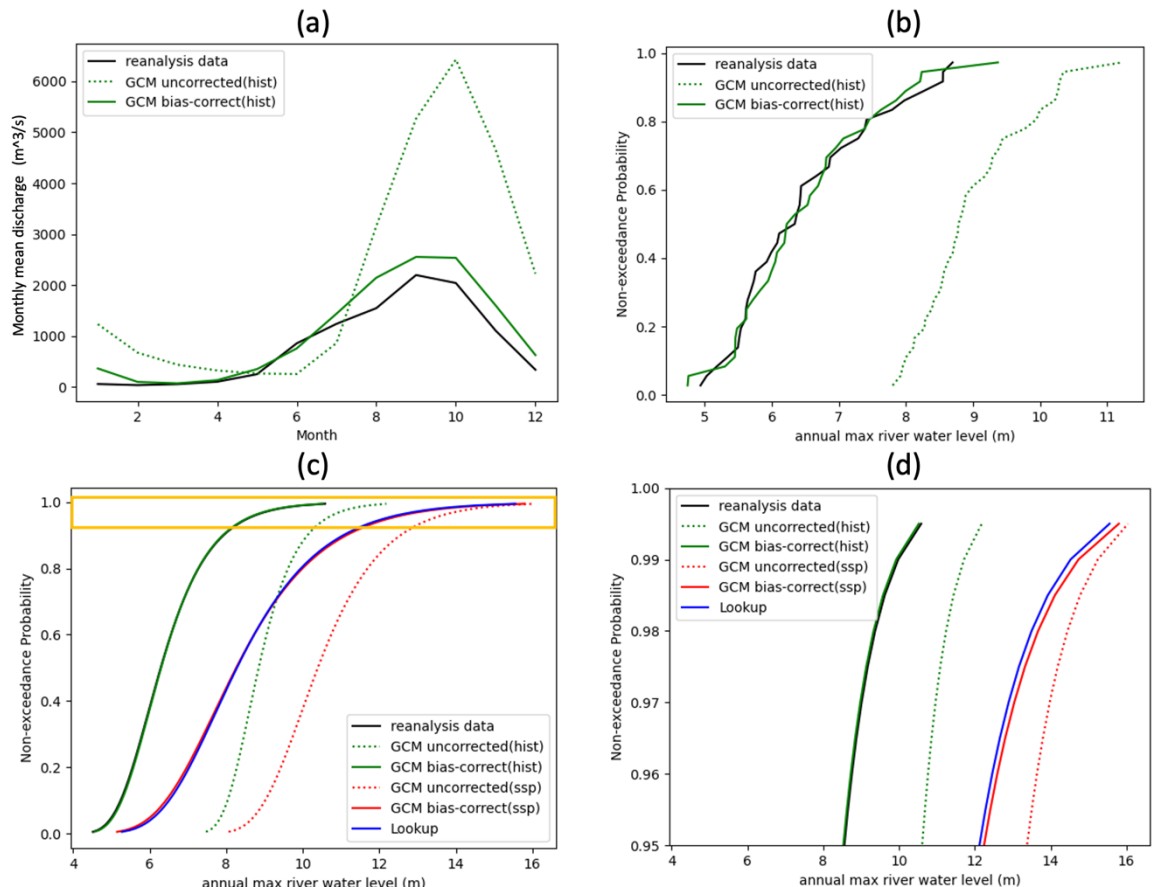


**Figure 5: Comparison of reanalysis data, uncorrected GCM, bias-corrected GCM and the lookup method at GRDC Ubon station. (104.8617°E, 15.2217°N). (a) Climatology of monthly mean discharge, (b) Exceedance probability curve of the annual maximum river water levels from 1980 to 2014, (c) Gumbel distribution of annual maximum river water levels, and (d) enlarged view of the orange square in (c).**





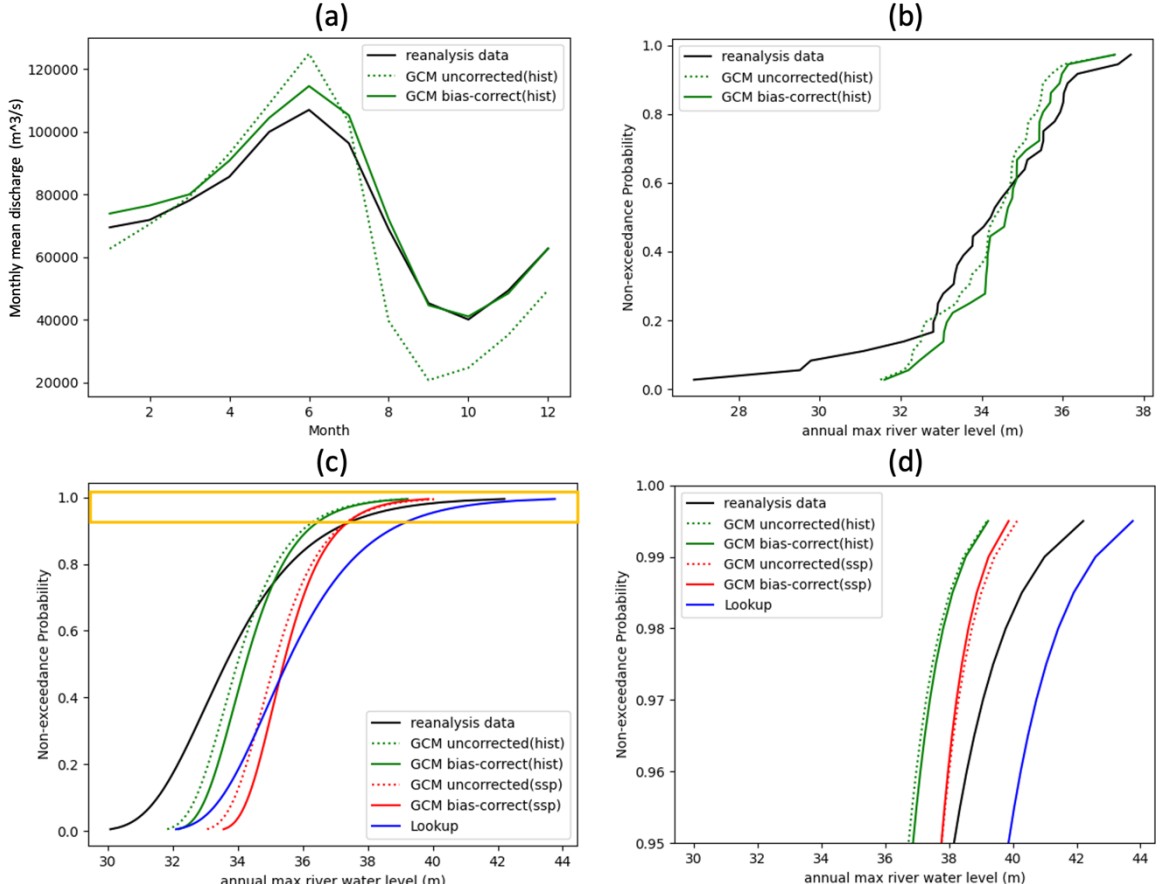

**Figure 6: Comparison of reanalysis data, uncorrected GCM, bias-corrected GCM and the lookup method at GRDC Itapeua station (63.0278°W, 4.0578°S). (a) Climatology of monthly mean discharge, (b) Exceedance probability curve of annual maximum river water levels from 1980 to 2014, (c) Gumbel distribution of annual maximum river water levels, and (d) enlarged view of the orange square in (c).**

Following analysis of the Chi-Mun River, a similar comparison was conducted for the Amazon River. Figure 6(a) shows that the climatology of monthly mean discharge of the bias-corrected GCM (historical) was similar to the reanalysis data, which was also as expected. On the other hand, the exceedance probability curve and Gumbel distribution of the annual maximum river water levels (Figure 6(b)–(d)) changed little with bias correction. The tail portion of the bias-corrected GCM (historical) was undervalued compared to the reanalysis data, implying that bias correction did not function as expected for extreme values associated with the hazard map, even under the historical climate. The runoff-correction method corrected for bias in GCM runoff data using monthly climatology based on reanalysis runoff values. However, as demonstrated by the Gumbel distributions in Figure 6(c), the bias of the GCM differed with the return period, indicating that the bias of extreme





values was not the same as that of the mean value. If the bias differs sharply among return periods, the runoff-correction method may not perform as expected for extreme values. The tail portion of the bias-corrected GCM (future) was also smaller than the tail for reanalysis data (Figure 6(d)), implying that direction of the change in future hazard obtained using the runoff-correction method relative to the reference reanalysis-based hazard map were inconsistent with the changes

projected by the GCMs. On the other hand, as shown in Figure 6(c)–(d), the lookup method accounted for changes in flood risk under the future climate projected by the GCMs. As described in Section 2.3.2, the lookup method uses the relative change in RP among GCMs, and can modify extreme values accordingly.

We recognized that bias correction of runoff using monthly climatology based on reanalysis runoff data produced large differences in the hazard map compared to the map based on reanalysis runoff. These differences arose because the bias of

extreme values differed from that of the mean value, implying that this bias correction procedure may not be suitable for extreme values on the hazard map. For extreme values of GCM runoff and river water levels, the results were not reliable even after bias correction, and such values may not be suitable for constructing hazard maps.

**3.2 Analysis of hazard maps generated with multiple GCM runoff datasets**

In this section, we compare the inundation areas of future-hazard maps constructed using two methods with nine GCMs of

CMIP6. The target area was the Chao Phraya River basin and surrounding rivers (98°E to 103°E, 12°N to 18°N). As shown in Figure 2, the direction of change from the reference historical hazard map was the same with both methods for Chao Phraya River basin and surrounding rivers when IPSL-CM6A-LR was used. In this section, we used other GCM models to test whether the two methods could produce hazard maps consistent with the future changes in flood risks predicted using the uncorrected GCM. Here, we compare the inundation areas of the 100-year RP hazard map among nine CMIP6 GCMs

(Figure 7).







### Inundation Area(km²)

**Figure 7: Box plot of inundated area (km²) estimates near the Chao Phraya River obtained using the runoff-correction method and lookup method. Whiskers show minimum and maximum values, boxes show 25th and 75th percentile values, orange lines show the median (50th percentile) value and green triangles show the average value.**

First, we examined whether the runoff-correction method for the historical period provided results similar to the reference reanalysis-based historical hazard map. The multi-model average inundated area for the nine models shown in Figure 7 indicates that the inundated area for the historical period was corrected from 45,124 km² to 45,508 km² using the runoff-correction method, and thus remained almost unchanged. Comparing the effects of bias correction for each model (Table S1 and Figure S1), we observed that MPI-ESM1-2 became closer to the reference historical hazard map than the uncorrected



GCM by using the runoff-correction method, while MIROC6 and GFDL-CM were corrected away from the reference values. These results imply that the inundation areas under the historical climate were not corrected as expected by the runoff-correction method for most GCMs. One reason for this finding is that, as noted above for the Amazon River (Figure 6), the bias-correction process did not properly correct for the Gumbel distribution of river water levels because the bias of extreme values was not the same as that of the mean values, and therefore bias correction did not address the extreme values on the hazard map. As shown in Figure 7 and Table S1, for the Chao Phraya River basin and surrounding rivers, the effect of bias correction was also small under the future climate in many models (using the runoff-correction method, the inundated areas for nine GCMs were corrected from 35,054–60,000 km$^2$ to 40,085–59,320 km$^2$). For example, as shown in Table S1, when using MPI-ESM1-2-HR, despite the increase in inundation area from the historical to future period for the uncorrected GCM, the inundation area of 40,085 km$^2$ obtained with the runoff-correction method was smaller than the corresponding area on the reference historical hazard map (42,321 km$^2$), implying that bias correction did not improve the GCM hazard map as expected.

The results of the lookup method were also examined. The lookup method is relatively unaffected by bias in the GCM, as the rate of change from the historical period to the future period is the key factor. If an inundated area increases from the historical climate to the future climate based on the uncorrected GCM, the inundated area should be larger on a hazard map produced using the lookup method than on the reference historical hazard map. Table S1 shows that the inundation areas obtained using the lookup method were consistent with the historical and future inundation changes of the uncorrected GCM. For example, using INM-CM5, the inundation area decreased between the historical and future periods based on the uncorrected GCM, implying a decrease in future flood risk. As shown in Table S1 and Figure S2, the inundation area was smaller for INM-CM5 using the lookup method than on the reference historical hazard map, indicating that the changes in future risk predicted by the GCMs were properly considered.

The results from multiple GCMs implied the following points of discussion. For most of the nine GCM models, the runoff-correction method did not correct the 100-year RP inundation areas under the historical climate as expected, and some GCMs were corrected away from the reanalysis data. Therefore, GCM-specific biases are unlikely to have been corrected on these hazard maps for the future climate. For example, the results from MPI-ESM1-2-HR implied that the runoff-correction method may not account for future climate risk projected by the GCM in terms of the direction of change from the reference historical hazard map. On the other hand, the lookup method did produce hazard maps consistent with the changes in flood risk under the future climate projected by the GCMs. Regarding the uncertainties in future flood-risk predictions, the divergence of the runoff-correction method includes the divergence of multiple GCMs in the future as well as the divergence associated with the bias of extreme values relative to that of the mean value in each model. The latter divergence reflects the model structure, specifically the systematic bias within each model. The lookup method removes this bias, and therefore it is expected to have smaller uncertainties than the runoff-correction method.





## 4 Discussions

### 4.1 Which method is more convenient for generating future-hazard maps?

In this study, we compared the runoff-correction and lookup methods, and investigated the causes of differences in hazard maps constructed using the two methods. Based on the results, we suggest using the lookup method with reanalysis data as a reference for the following three reasons.

First, using a hazard map based on reanalysis runoff data as reference is convenient. Reanalysis runoff values are based on historical weather observations, and thus a hazard map created using reanalysis runoff is expected to show good consistency with actual flood risk. The performance of global river model simulations using reanalysis runoff values has been evaluated through comparison with observed data (e.g., time series of discharge, water level, and inundation extent; Yamazaki et al., 2011, 2014). Hazard maps generated through global river model simulations with reanalysis runoff have been validated in numerous studies. Bernhofen et al., 2018 validated several global flood models, including CaMa-Flood, which was used in this study. The accuracy of hazard maps produced using global flood models has been validated through comparisons with existing flood-hazard maps (e.g., CaMa-Flood results in Japan were validated by Kita et al., 2022, and GloFAS model results were validated in Europe and the Mediterranean by Dottori et al., 2022).

Second, the lookup method can produce hazard maps that are consistent with projected changes in future flood risk based on GCMs, as demonstrated by the results of this study, discussed in Section 3. On the other hand, direction of the change in future hazard (increase or decrease) obtained using the runoff-correction method relative to the reference reanalysis-based hazard map may be inconsistent with the changes projected by the GCMs, as described for the Amazon River in Section 3. The lookup method also has the advantage of facilitating research on efficient construction of future climate hazard maps, as it allows for improvement of the reanalysis hazard map by upgrading the model, and the estimated changes due to climate change can be considered separately.

Third, the use of GCM historical climate simulations as reference data introduces problems. As noted in Section 3.2, for most of the nine GCM models, the runoff-correction method did not correct the 100-year RP inundation area as expected based on the historical climate, and some GCMs were corrected away from the reanalysis data, indicating that the accuracy of bias-corrected GCM historical climate data is poor. In addition to evaluating the future climate hazard map itself, many situations require the future-hazard map to be considered in comparison with a historical climate hazard map (Hirabayashi et al., 2021 and Taguchi et al., 2022). Using reanalysis data to construct the hazard map for the historical climate makes such analyses easier. However, when GCMs are used for the historical climate, the inundation area of historical climate is calculated individually for each GCM; in such cases, multiple results are referenced, which complicates mapping and comparison.

For these reasons, we consider that the lookup method with reanalysis data as reference data is the most reasonable method of creating future-hazard maps for flood risk assessment.




## 4.2 Validity of reverse slope revision

As noted in Section 2.2, we revised the reverse slope during the process of hazard-map generation in this study. For actual physical processes, if slope reversal occurs, the backwater effect causes the downstream water level to propagate upstream (backflow), which increases the flood risk. A reverse slope is a technical problem in extreme-value analysis. In the process

of calculating the water levels for the targeted return period, we fitted a Gumbel distribution to each grid, allowing the upstream water surface elevation to be lower than the downstream elevation, resulting in an unrealistic reverse water slope. If such a reverse slope is not revised and the backwater effect is not considered, the inundation depth distribution may not be physically reasonable and flood risk may be underestimated.

Therefore, we analyzed the effect of revising the reverse slope and present our findings in this section (reverse-slope

revision is referred to as Backwater_Modification in this section). Prior to analysis of the Backwater_Modification effect, the occurrence of slope reversal was checked using the water surface elevation distributions from the reanalysis data (without Backwater_Modification) and the lookup method (without Backwater_Modification). The water surface elevation distribution obtained through application of extreme-value analysis to each grid water level showed that many reverse slopes occurred, although the majority were small (with 6-arcmin global resolution, 30,000 grids for the reanalysis data and 37,000

grids for the lookup method). Reverse-slope grids occurred frequently at confluences, where the backwater effect can occur, and thus reverse slope revision is physically reasonable.

We applied Backwater_Modification to the water surface elevation and assessed the impact of this modification. The area of 83.5°E to 86°E and 25°N to 26.5°N, where many reverse slopes are shown in Figure S3(b), was the target area for this process. The water surface distribution was created by adding elevation to the 100-year RP inundation depth distribution. As

shown in Figure 8(a)–(b), the reverse slope was eliminated by Backwater_Modification. In addition to checking the water surface distribution, we drew the water surface on a cross-section of the river and a section from upstream to downstream (red and green lines in Figure 8) to check whether the water surface was smoothly revised. Figure 8(d) shows that the water surface in the cross-section of the river tended to change significantly at the boundary of the unit catchment without Backwater_Modification; however, with Backwater_Modification the water surface distribution was smooth. In addition, as

shown in Figure 8(e), for the water surface section from upstream to downstream, the reverse-slope condition was resolved, leading to a water surface distribution that was physically reasonable. As shown in Text S1, additional validation of Backwater_Modification was conducted. Specifically, we compared the CaMa-Flood-hazard maps obtained using Backwater_Modification with hazard maps published in Japan containing information about inundation depths. The comparison results (Table S2 and Figure S4) showed that the inundation area was more realistic with

Backwater_Modification than without it. As noted above, if a reverse slope is present in the water surface level, revision using the method proposed in this study (Backwater_Modification) would be appropriate.





**Figure 8: (a) Water surface distribution obtained using the lookup method without Backwater_Modification; (b) water surface distribution obtained using the lookup method with Backwater_Modification; (c) the difference in water surface level between (a) and (b); (d) water surface profiles along the red transects (cross-section of the river); and (e) water surface profiles along the green transects (section along the flow direction (from upstream to downstream))**

## 4.3 Future changes in inundation area and the population affected within the inundation area

We analyzed the extent of differences in the estimates of future flood risk changes, specifically inundation area and the affected population within the inundation area depending on the implementation of bias correction or Backwater_Modification and the method used to construct the hazard map. The flood-exposed population was estimated based on the inundation map and the 2020 population density map (Gridded Population of the World; CIESIN, 2018). This



map has 30-arcsec resolution, and therefore the 3-arcsec inundation map was aggregated to 30-arcsec resolution. Here, as in Section 3, we present the results from the 100-year RP hazard map of IPSL-CM6A-LR, which is one of the nine CMIP6 GCM models. Based on the inundation area in Table 2, we found that the 100-year RP inundation area was 14.0 million km$^2$ based on reanalysis runoff and 20.3 million km$^2$ based on uncorrected GCM runoff (future), indicating an increase of approximately 45% compared to the historical climate. On the other hand, the 100-year RP inundation area reached 20.5

million km$^2$ using the runoff-correction method (relative to historical climate: +46%) and 19.1 million km$^2$ using the lookup method (relative to historical climate: +37%); thus, the difference in future flood risk estimates was up to 9%, depending on whether bias correction was applied and the construction method. Inundation areas with particularly high risk of inundation depths of 5 m or greater covered 1.3 km$^2$ in the historical reference period and ranged from 3.0–4.2 km$^2$ under the future climate (relative to historical climate: +124%–216%) for various methods, indicating a difference in future flood-risk

estimates of up to 92%, especially in high-risk areas. The effects of Backwater_Modification were then analyzed. Table 2 shows that it increased the inundation area from 18.98 million km$^2$ to 19.13 million km$^2$ for the lookup method, an increase of about 0.15 million km$^2$. Focusing on high-risk areas, the inundation area based on the lookup method increased from 2.85 million km$^2$ to 2.98 million km$^2$ with Backwater_Modification, an increase of approximately 0.15 million km$^2$ in high-risk areas. Notably, without Backwater_Modification, the overall inundated area was underestimated by about 0.15 million km$^2$,

and the high-risk area was also underestimated by about 0.15 million km$^2$. Table 2 shows that the affected population tended to be similar to the population of the inundated area.

**Table 2: Inundation area of the 100-year RP hazard maps and affected population from the 100-year RP hazard maps constructed using various methods; "_no_modification" indicates a map constructed without Backwater_Modification. These results were**
**obtained using IPSL-CM6A-LR.**

|  |  | Inundation Area (Unit: Million km^2) | | Exposed Population (Unit: Million) | |
|---|---|---|---|---|---|
|  |  | all | depth > 5m | all | depth > 5m |
| Historical Climate | Reference historical (Reanalysis) | 14.00 | 1.33 | 1,269 | 158 |
| Future Climate | GCM uncorrected | 20.28 | 4.20 | 1,783 | 395 |
|  | Runoff Correction Method | 20.45 | 4.02 | 1,802 | 406 |
|  | Lookup_Method_no_modification | 18.98 | 2.85 | 1,755 | 352 |
|  | Lookup_Method | 19.13 | 2.98 | 1,784 | 367 |

     An alternative method for risk assessment under a future climate in areas where future-hazard maps have not been created is the use of hazard maps for historical climates to estimate future damage. We considered a method of future risk

assessment that does not use future-hazard maps, i.e., the use of hazard maps created for historical climates and calculation of the exposed population based solely on future frequency changes. As shown in Table 2, the areas inundated by a 100-year RP flood in the future climate will differ from those inundated under the historical climate. Therefore, properly assessing flood risk under a future climate is not possible if the hazard map is fixed and only changes in frequency are considered.





Table 2 indicates the total population living in the inundation areas modeled for the reference historical hazard map to be
around 1.27 billion. Using the fixed hazard map method, the affected population in the future climate would also be 1.27
billion. However, population estimates of 1.80 billion with the runoff-correction method and 1.78 billion with the lookup
method were obtained in this study, implying that an affected population of around 0.5 billion may be overlooked if the
hazard map is fixed and only changes in frequency are considered. As a specific example, we identified the affected
population in the Chao Phraya River and Mekong River area, and Figure 9 shows the affected population in that area based
on the 100-year RP hazard map. In that area, the total affected population based on the reference historical hazard map is 63
million and the population obtained for the future-hazard map using the lookup method is 79 million, indicating
underestimation by approximately 16 million if the hazard map is fixed and only frequency change is considered (Figure
9(c)).

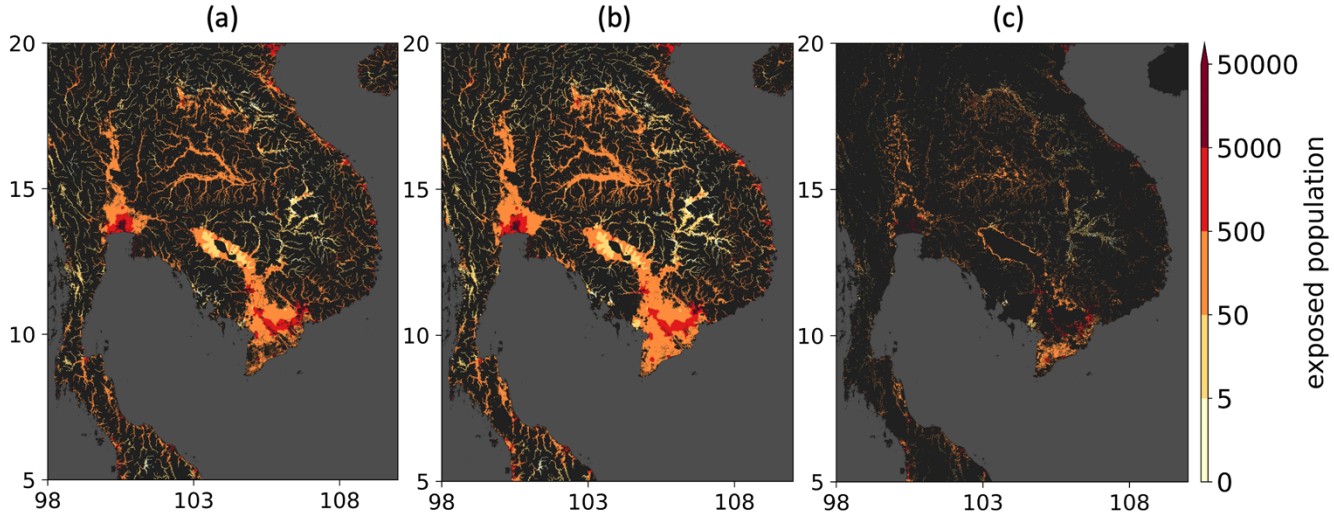

**Figure 9: Affected population in the Chao Phraya River and Mekong River area based on the 100-year RP hazard map (obtained using IPSL-CM6A-LR). (a) Reference historical hazard map; (b) future-hazard map constructed using the lookup method; and (c) difference of (b)−(a), representing the population not currently affected by flooding that may be affected under the future climate.**

**5 Conclusions**

In this study, we explored several methods for constructing hazard maps under a future climate, including bias correction,
and investigated which of these methods could produce a reasonable inundation depth distribution. The results imply that
bias correction of runoff using monthly climatological data based on reanalysis runoff values produced a significant
difference in the hazard map compared to the map based directly on reanalysis runoff values, as the bias of extreme values
was not the same as that of the mean value. In addition, we found that the direction of the change in future hazard (increase



or decrease) obtained using the runoff-correction method relative to the reference reanalysis-based hazard map may be inconsistent with the changes projected by the GCMs. On the other hand, we confirmed that the lookup method, which uses the statistical frequency of flooding calculated by the GCMs and the reanalysis data for inundation depths corresponding to that frequency, could produce hazard maps consistent with the changes in flood risk projected by the GCMs, indicating the

possibility of obtaining a reasonable inundation area distribution.

In addition, we discussed which method is most convenient for generating future-hazard maps in terms of ease of use. When using the GCM historical climate with bias correction based on the climatology of average monthly runoff, the inundation areas were calculated separately for each GCM. For the Chao Phraya River, as described in Section 3, the 100-year RP inundation area under the historical climate varied widely, from 34,475 to 54,121 km$^2$, making it difficult to use in

practice. In contrast, the lookup method defines the historical climate hazard map as having a value of one, making analysis simpler. Therefore, the lookup method is superior in terms of ease of comparison with reanalysis data. Regarding the uncertainties in future trends, as described in Section 3.2, the lookup method removed the systematic bias in the distribution of annual maximum water levels within each model, and therefore is expected to have smaller uncertainties than the runoff-correction method. The lookup method also has the advantage of facilitating efficient construction of future climate hazard

maps, as it allows for separate consideration of reanalysis hazard map improvement through upgrading of the model and estimation of changes associated with climate change. Our results imply that hazard maps could be made more realistic by applying the proposed method to revise water slope reversal. Based on these findings, we suggest use of the lookup method with reanalysis data as a reference.

In addition, this study examined the extent to which estimates of future flood risk changes differed depending on whether

bias correction was implemented and the method used for future-hazard mapping. Our assessment of inundation areas and the future population affected by flooding showed that the variation in future flood risk estimates due to bias correction and the method of model construction was up to 9%. Using the lookup method, the total population living in the modeled inundation areas where the flood magnitude exceeded the 100-year RP under a future climate was estimated to be around 1.8 billion. In addition, our risk assessment under a future climate showed underestimation of around 0.5 billion for the affected

population when the historical hazard map was used as an alternative to future-hazard maps and only the change in frequency was considered. These results imply that global flood risk studies require future-hazard maps, i.e., inundation depth distributions at high resolution, for proper estimation of climate-change risk, and that discussing only changes in the frequency of a given flood intensity is insufficient.

**Code availability**

The global hydrodynamic model CaMa-Flood (v4.01) is available from (https://doi.org/10.5281/zenodo.4659583) (Yamazaki et al,. 2021).



**Data availability**

The topography data MERIT are available from http://hydro.iis.u-tokyo.ac.jp/~yamadai/MERIT_DEM/index.html (last access: 17 October 2022) (Yamazaki et al., 2019). VIC-Bias-Corrected runoff are available from
https://www.reachhydro.org/home/records/grfr (Yang et al., 2021). The CMIP6 data are available from the Earth System Grid Federation (ESGF) data platform (https://esgf-node.llnl.gov/search/cmip6/, accessed on 17 October 2022).

**Author contribution**

Yuki Kimura, Yukiko Hirabayashi and Dai Yamazaki conceived the study and contributed to the development and design of the methodology. Yuki Kimura performed simulation and analysis. Xudong Zhou developed a code for extreme value
analysis. Yuki Kimura and Dai Yamazaki prepared the manuscript with re-view from Yukiko Hirabayashi, Xudong Zhou, Yuki Kita.

**Acknowledgements**

This research was supported by LaRC-Flood Project co-funded by MS&AD and NEDO (JP21500379)




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
