# Peer review of "Methodology for constructing a flood-hazard map for a future climate"

_EGUsphere, 2022_

## Author Comment (AC2)

**In this document, the review comments are in black, our responses is in red and the revised text are in blue.**

This paper discusses the generation of future hazard maps in a global inundation modelling context. As present day global hazard models have become more common and more detailed interest has been building in creating future hazard maps from these data. This study is a timely analysis examining some of the decisions involved in propagating GCM analysis into these hazard data sets and thus a valuable addition to the literature.

Overall the analysis has been done well and the paper is clearly written with an easy to follow structure. The novelty of the work could be more precisely defined in the induction because many local scale climate conditions hazard maps have been created in the past (see specific comments below). Furthermore, although I appreciate why the authors have developed a method to ensure monotonically decreasing water levels, I don't think the rational for doing this (backwater effect) is a clear cut as suggested in the text because a hazard map is not a physically coherent event. If the authors agree with my comments then they should be able to modify the arguments in the text and publish the research.

Reply:

We would like to express our gratitude to referee #1. We will address all the comments in the revised manuscript, and comprehensive explanations are provided below.

Backwater_Modificiation is applied in order to revise the spatial inconsistency that is due to the distribution fitting (fitting Gumbel distribution) at each grid (unit-catchment). It is possible that it would be overestimated by applying Backwater_Modification because they are corrected at all reverse slope occurrence points. Reversed water surface slopes might exist in conventional flood hazard maps, given that they are not always constructed by one flood event simulation as pointed out by referee #1.

In light of the above perspectives, we have conducted validation to investigate whether Backwater_Modification improves the consistency to national hazard maps or not. In result, we have found that the Backwater_Modification brought us closer to the hazard map prepared by Japanese goverment, indicating that it is important to apply Backwater_Modification.

Specific points:

Line 58: It would be good to add some references on in country hazard mapping. In general as this paragraph is an overview of local-scale flood hazard mapping efforts I found it a bit citation lite. It would be nice to signpost some other review articles as De Moel et al 2009 was published some time ago.

At the risk of self-promotion (no need to include any of this unless it's useful to support your later arguments) we included a UK focused review of local-scale modelling efforts that highlights some of the challenges of using these data sets in a climate conditioned application.

Bates, P. D., Savage, J., Wing, O., Quinn, N., Sampson, C., Neal, J., and Smith, A.: A climate-conditioned catastrophe risk model for UK flooding, EGUsphere [preprint], https://doi.org/10.5194/egusphere-2022-829, 2022.

Reply: We agree with referee #1 suggestion, so we will add "Bates et al., 2022" and "Wing et al 2022" as a reference.

Bates, P. D., Savage, J., Wing, O., Quinn, N., Sampson, C., Neal, J., and Smith, A.: A climate-conditioned catastrophe risk model for UK flooding, EGUsphere [preprint], https://doi.org/10.5194/egusphere-2022-829, 2022.

Wing, O. E., Lehman, W., Bates, P. D., Sampson, C. C., Quinn, N., Smith, A. M., Neal, J. C, Porter, J. R, and Kousky, C.: Inequitable patterns of US flood risk in the Anthropocene. Nature Climate Change, 12(2), 156-162, https://doi.org/10.1038/s41558-021-01265-6. 2022.

"In the research field, for example, Bates et al., 2022 estimated annual flood damage in UK and Wing et al., 2022 also estimated flood damage in US by using local-scale flood-hazard maps. "

"Line 81:assessment of future flood risks based on the spatial distribution of inundation depths has not yet been established" I agree at the global scale but there are many local scale examples of such studies that have had to consider the GCM bias issue. I think that needs to be acknowledged explicitly in this section or the previous paragraph.

Reply: We would like to express our gratitude to referee #1. We will revise the sentence as follows. We will clarify that in this paper we described a "global" study. We will also clarify that flood "risk" has been assessed, but flood "hazard" itself has not been sufficiently verified in previous global-scale studies.

"While assessment of climatic and meteorological hazard under climate change (e.g. extreme temperatures, droughts and heavy-rainfall events) has been widely performed using direct output variables of general circulation models (GCMs), such as precipitation and temperature (Li et al., 2021, Lu et al., 2019), at present, no global high-resolution flood hazard (i.e. inundation depth) in a future has been sufficiently verified. Even though some studies assessed future flood risks (e.g. affected population and GDP) at the global scale (Ward et al., 2020b), it is important to analyze global future flood-hazard (i.e., inundation depth distributions), and also important to assess uncertainties such as those caused by different bias corrections."

For example, this sentence "While bias-correction methods for precipitation and temperature have been studied in detail (Watanabe et al., 2012, Hempel et al., 2013 and Lafond et al., 2014), such methods have not been established for runoff data for use as inputs to flood models" is not correct without a more precise focus on the novelty of this study because there are many examples of localized climate conditions hydrological cascades all the way to inundation hazard and impact. These studies have some of the same issues you so I would have thought there were some useful conclusions from the literature that might feed into this section.

Reply: We will revise the sentence to clarify and focus on the novelty of this study. Specifically, we will change from "such methods have not been established for runoff data for use as inputs to flood models" into the following sentence.

"such methods have not been established for runoff data for use as inputs to **global** flood models to **construct large-domain** future flood **hazardmap.**"

Line 113: "Although CaMa-Flood is a global model, it is unique in that it represents the physical processes necessary to reproduce floodplain inundation dynamics" I appreciate there are un

ique aspects to CaMa-Flood, however there are several global flood inundation models so perhaps remove the word "unique" or be more specific about the combination of capabilities that are unique.

Reply: In response to referee #1 comments, we will remove "unique" and revise the sentence as follows.

"Although CaMa-Flood is a global model, it **has characteristics** that it represents the physical processes necessary to reproduce floodplain inundation dynamics."

Line 137: could you be specific about the simulation length – I suspect its shorter than most readers will assume by the wording long term. Is it 35 years?

Reply: As you understand, the simulation length is 35 years. Based on referee #1 comment, we will remove "long-term" and added "(time period: 1980-2014)" to the end of the sentence to specify about the simulation length.

"we conducted a historical river hydrodynamics simulation with a daily time step using observation-based runoff data (Reanalysis_Runoff) as an input to CaMa-Flood **(time period: 1980-2014)**."

Line 150: I have no issue with this pragmatic method for getting a spatially consistent hazard map, but I'm not convinced by the description. The hydrodynamic model simulated the backwater effect so I assume that this effect is captured in the hazard/level simulations. Could it be the case that the spatial inconstancy is therefore due to the distribution fitting and other data processing factors rather than a physical backwater effect being omitted? I appreciate the maps will look better with this correction but do you risk biasing the levels higher in the process? On a more fundamental level a hazard map is not an event, so does the water level even need to monotonically decrease downstream since you simple seek to best simulate the hazard rather than a plausible physical water surface which a hazard map will never be?

Reply: We thank referee #1 for comment. we will revise the sentence to make it clear that it is the case that the spatial inconstancy is due to the distribution fitting at each unit-catchment scale (grid-scale) as follows.

"If reverse-slope revision is not conducted, **reverse slope occurring through fitting of Gumbel distributions remain** and the inundation-depth distribution may not be physically reasonable. For this reason, a novel reverse-slope revision method was applied in this study **in the purpose of revising the spatial inconstancy that is due to the distribution fitting at each unit-catchment scale (grid-scale)**."

In response to referee #1 comment " I appreciate the maps will look better with this correction but do you risk biasing the levels higher in the process? ", we will add the notes of overestimation to the last paragreaph in section 4.2 (Specifically, the following sentence will be added).

"Please note that it is possible that it would be overestimated by applying Backwater_Modification because they are corrected at all reverse slope occurrence points. Reversed water surface slopes might exist in conventional flood hazard maps, given that they are not always constructed by one flood event simulation. In light of the above perspectives, we have conducted validation to investigate whether Backwater_Modification improves the consistency to national hazard maps or not."

Section 2.3.1: Could you cite some alternatives to the bias correction method and explain why this one was chosen. I'm not an expert in these methods, but my assumption is that the results of this method would be sensitive to the choice of bias correction. Depending on your response above you might need to caveat the conclusions drawn around line 336.

Reply: We thank referee #1 for comment. We will cite some alternatives to the bias correction method and explain why this one was chosen.

"As an alternative to addictive correction method, there are multiplicative correction method, which multiplies the ratio with the reanalysis data, and Quantile Based Mapping method (Panofsky and Brier 1968, Watanabe 2020), which is to obtain ordinal statistics from the reanalysis data and the GCM, and to create an equation relating these statistics. In the future cl

imate, the average monthly runoff may fluctuate significantly due to changes in the arid zones caused and changes in the timing of the wet and dry seasons by increasing temperatures. Based on the above, this study used addictive correction method, which is relatively insensitive to the above fluctuations."

Did you consider looking at more frequent hazards? I assume the runoff bias correction method would work better the closer you were to the mean annual flood because the bias would be better corrected? Or maybe the distribution fitting is still impacted…

Reply: We looked at more frequent hazards, the runoff bias correction method worked closer to reanalysis data than RP100 in some cases. For example in GRDC BAN_BANG_KAEO station (100.4533°E, 14.5847°N), as below figures are seen, the runoff bias correction method worked closer to reanalysis data at more frequent hazards (e.g. RP2-10) than RP100 in cases.

[Figure]

Figure : Comparison of reanalysis data, uncorrected GCM, bias-corrected GCM at GRDC BAN_BANG_KAEO station (100.4533° E, 14.5847° N). (a) Gumbel distribution of annual maximum river water levels and (b) enlarged view of the orange square in (a).

Line 383: "On the other hand, the lookup method did produce hazard maps consistent with the changes in flood risk under the future climate projected by the GCMs." I found this a bit confusing. Do you mean flood risk here? I assume you can only derive risk from the hazard maps, do you mean the changes in GCM runoff or precipitation? I think this section needs to be more precise about what you are looking for constancy with because I don't think its risk – or at least I didn't understand how it could be from the text.

Reply:

As you pointed out, because we use "flood risk" uncorrectrly in the sentence, we will revise the sentence to clarify "the changes in flood **hazard** under the future climate projected **by C aMa-Flood simulations**"

Specifically, we will change from "the changes in flood risk under the future climate projecte d by the GCMs" into as follows.

"the changes in flood **hazard** under the future climate projected **by CaMa-Flood simulatio ns with input of GCMs runoff**"

Line 390: "4.1 Which method is more convenient for generating future-hazard maps?" Conven ient is the wrong word here it only applies to some of your arguments.

Reply: We agree with your comment. we will change from "convenient" into "reasonable" in the title of section 4.1.

"4.1 Which method is more **reasonable** for generating future-hazard maps?"

Line 394: I think you could also note here that this approach can also be used with hard mapp ing methods based on regional flood frequency analysis or machine learning from gauging stat ion data, which at least in historically well monitored river reaches can be more accurate than rainfall-runoff reanalysis based methods (perhaps this is contested) for present day hazard ma pping and extreme flow estimation because the modelling chain is much shorter (e.g. Laura D evitt et al 2021 Environ. Res. Lett. 16 064013 DOI 10.1088/1748-9326/abfac4)

Reply:

Thank you very much for your suggestion.

As you pointed out, there is no need to limit it to reanalysis-basis as a "historical hazard ma p". Based on the above, we have added the following sentence to the the end of 1st paragr aph in section 5 "Conclusion".

" This implies that combining accurate historical hazard maps with information on future fre quency changes of floods is considered optimal in general for generating future hazard map. Please note that the historical flood hazard map does not have to be reanalysis-based simul ation using GFM, and the proposed method can be also applicable to gauge-based or machi ne-learning based historical hazard map."

Line 407 I think this is a separate point from the one above "The lookup method also has the advantage of facilitating research on efficient construction of future climate hazard maps, as it allows for improvement of the reanalysis hazard map by upgrading the model, and the estima ted changes due to climate change can be considered separately" and you could expand to dis cuss using multiple flood hazard models for the analysis as several are introduced in the prece ding section.

Reply: We would like to express our gratitude to referee #1 suggestion. We will add one poi nt to advantage of the the lookup method

The lookup method also has the advantage of facilitating research on efficient construction o f future climate hazard maps because preparation of historical hazard map and estimation of future frequency change can be separated. This is beneficial for two aspects: 1) it allows for improvement of the reanalysis hazard map by upgrading the model; **2) it allows for use of multiple reference hazard maps by using different reanalys-based simulations.**

Section 4.2: I think it's valuable to discuss this correction (so I'm not suggesting a major revis ion). But in my opinion it's debatable whether this correction is desirable for a hazard map whi ch is not an event and thus not a physically possible water surface. I think it potentially double counts for the backwater effect and over-predicts the flood inundation levels by biasing locatio ns upstream to any over-prediction errors in the distribution fitting downstream. That said the hazard mapping is improved relative to some high quality validation data so I don't dispute th at some form of postprocessing of the levels to aid spatial constancy is the wrong thing to do ⋯ but you could smooth for example. Personally I would slightly modify the discussion in secti on 4.3 to present the results with and without the backwater modification as an indicator of th

e sensitivity to this issue – which is less than the climate change signal and impact of different GCM treatments. This would also require a small edit to the conclusion ~line 532.

Reply: We would like to express our gratitude to referee #1.

Backwater_Modificiation is applied in order to revise the spatial inconsistency that is due to the distribution fitting(fitting Gumbel distribution) at the unit-catchment scale. In response to referee #1 comment "I think it potentially double counts for the backwater effect and over-predicts the flood inundation levels by biasing locations upstream to any over-prediction errors in the distribution fitting downstream." we will add the notes of overestimation (Specifically, the following sentence will be added) in the last paragraph on section 4.2.

"Please note that it is possible that it would be overestimated by applying Backwater_Modification because they are corrected at all reverse slope occurrence points. Reversed water surface slopes might exist in conventional flood hazard maps, given that they are not always constructed by one flood event simulation. In light of the above perspectives, we have conducted validation to investigate whether Backwater_Modification improves the consistency to national hazard maps or not."

Line 465: why choose lower resolution population data rather than say Worldpop? Could this bias the results?

Reply: We would like to express our gratitude to referee #1 question.

In this study, as to the estimates of future flood risk change, we compared the differences between the two methods to construct a future hazard map or with/without implementation of bias correction. Since the purpose is **to analyze the impact of correction at the global scale**, we think that 30 arcsec resolution was enough.

We used Gridded Population of the World; (CIESIN, 2018), which has data that are stored together for the entire world, rather than Worldpop, whose data are broken down by country.

In addition to above, we will add the following notes regarding uncertainty in spatial resolution in section 4.3.

"As per Zhou et al., 2021 discussed, the spatial resolution of the flood hazard map is a particularly important determiner of impact assessment. Smith et al. (2019) evaluated the popul

ation exposure to a 1-in-100 year flood in 18 developing countries, showed that decreasing the spatial resolution of flood hazard map from 90 to 900 m increases the exposure by 5 1 % to 94 % for different population products. Although there is uncertainty involved in the choice of the spatial resolution of the flood hazard map as per above, we used 30 arcsec res olution instead of 3arcsec resolution for the purpose of comparison between methods to con struct a future flood hazardmap on a global scale in this section."

---

## Author Comment (AC3)

**In this document, the review comments are in black, our responses is in red and the revised text are in blue.**

*This paper analyzes two methods for the generation of future flood hazard maps under climate change, along with a number of related issues. Even though the work focuses on a global-scale application of the CamaFLood model, the outcomes are relevant from a general point of view, given the importance of the topic. Overall the paper is well structured and generally well written, although some descriptions could be improved (see my comments). The analyses carried out are appropriate and well described. I think that the paper could be published after addressing a few minor issues that I am listing below.*

**Reply: We would like to thank the referee for his kind remarks. We will address all the comments in the revised manuscript, and comprehensive explanations are provided below.**

*L25: "...changes in flood risk..." I would correct in "changes in flood hazard"*

**Reply: As you pointed out, because we use flood risk incorrectrly in the sentence, so we will revise it from risk to hazard.**

**"On the other hand, we confirmed that the lookup method can produce future-hazard maps that are consistent with the changes in flood hazard projected by CaMa-Flood simulations with input of GCMs runoff, indicating the possibility of obtaining reasonable inundation-area distribution."**

*L29 "we discuss future changes at global scale..."*

**Reply: We will revise the sentence as referee #2 suggested.**

*L49-50: "To elucidate the potential impacts of flood disasters, a high-resolution map of potential disaster impacts must be developed, commonly named a hazard map." This is not fully correct. In flood risk literature, hazard is a component of risk but it is not a synonym of disaster impact (see for instance Ward et al, 2020). Perhaps you could replace with "To elucidate the potential impacts of flood disasters, high-resolution maps of disaster impacts must be developed".*

**Reply: In order to make an appropriate sentence that leads to the commonly named a hazard map, we will revise the sentence as referee #2 suggested. We will use the term "high-resolution inundation-depth maps"**

*L70-72 "Uses of large-domain flood-hazard maps include estimation of the affected population within an inundation area and determination of the impacts of flooding on GDP and urban areas in the current climate". This sentence should be reworded. In literature, the maps including impacts such as population and/or urban areas exposed arew generally called flood risk maps*

**As you pointed out, the sentence was not clear and was not stated correctly, so we will revise as follows to clarify the meaning of "Large-domain flood-hazard maps were used for many purposes such as estimation of risks." Specifically, we will revise it as follows.**

**"Large-domain flood-hazard maps was used for many purposes such as estimation of the affected population within an inundation area and determination of the impacts of flooding on GDP and urban areas in the current climate (Ward et al., 2020a)."**

*L72-74: Bernhofen et al (2018) compared six global flood models against satellite-derived flood maps, so adding a reference here would be appropriate in my view.*

**Reply: We agree with your point. We will add Bernhofen et al (2018) to the sentence as referee #2 suggested.**

*L80-81 this sentence should be modified, because several flood risk assessments have been carried out at scales from global to local*

**Reply: We will revise the sentence as follows. We will clarify that in this paper we described a global study. Also, we will clarify that flood risk has been studied, but flood hazard itself has not been sufficiently verified.**

**"While assessment of climatic and meteorological hazard under climate change (e.g. extreme temperatures, droughts and heavy-rainfall events) has been widely performed using direct output variables of general circulation models (GCMs), such as precipitation and temperature (Li et al., 2021, Lu et al., 2019), at present, no global high-resolution flood hazard (i.e. inundation depth) in a future has been sufficiently verified. Even though some studies assessed future flood risks (e.g. affected population and GDP) at the global scale (Ward et al., 2020b), it is important to analyze global future flood-hazard (i.e., inundation depth distributions), and also important to assess uncertainties such as those caused by different bias corrections."**

*L147-154: I think that the description of the post-processing method needs more detailing. In particular I have some questions:*

*- My understanding is that the authors fitted a Gumbel distribution on each pixel of the Camaflood 6-arcmin grid, correct? Or do you use different areas for the fitting?*

**Reply: We will revise the sentence as follows to clarify that fitting a Gumbel distribution on each pixel of the Camaflood 6-arcmin grid.**

**"We fitted a Gumbel distribution on each of the Camaflood 6-arcmin grid."**

*- In lines 155-162 you state that water surface elevation is uniform within each 6-min unit catchment, so I assume that water level in upstream catchment are increased to the same water level of donwstream catchment, right? If yes, please specify this in the text*

**Reply: Referee #2's understanding is correct. As referee #2 suggested that, we will revise the sentence as follows.**

**"To avoid this issue, if a reverse water slope was obtained in the water surface elevation distribution, we elevated the water surface elevation of upstream catchments to match those of downstream catchments."**

*- Can you also explain why this approach was not needed in previous studies based on CamaFlood?*

**Reply: We thank referee #2 for question. We would say the risk estimate of the previous studies could be improved by applying this reversal-slope modification approach. As described in the introduction, flood depth distribution itself had not been extensively evaluated in previous studies.**

**In addition, the purpose of applying reverse-slope revision is written more clearly as follows.**

**"If reverse-slope revision is not conducted, reverse slope occurring through fitting of Gumbel distributions remain and the inundation-depth distribution may not be physically reasonable. For this reason, a novel reverse-slope revision method was applied in this study in the purpose of revising the spatial inconstancy that is due to the distribution fitting at each unit-catchment scale (grid-scale)."**

*L218: Alfieri et al (2017) actually employed the lookup method, because they used historical flood hazard maps coupled with changes in frequency under future climate scenarios.*

**Reply: We will add Alfieri et al (2017) as a reference as follows.**

"As the lookup method does not use GCM_Runoff_BC and thus avoids the uncertainties associated with bias correction, including questionable results of bias correction for extreme events (Alfieri et al., 2017; Huang et al., 2014), several previous studies have employed this technique (Hirabayashi et al., 2013; Hirabayashi et al., 2021). Alfieri et al., 2017 employed the lookup method, they estimated future affected population and damage by flood using historical flood hazard maps coupled with changes in frequency under future climate scenarios. "

*L405-406: Based on the results, the authors could maybe draw the conclusion that standard bias-correction techniques of GCM data are not suitable for use in flood hazard estimation (having been developed for different types of climate studies), and that different bias-correction techniques should be used (i.e. more focused on extreme values)*

**Reply: We agree with referee #2 suggestion. In response to referee #2' suggestion, we will revise the sentence in the section 4.1 as follows.**

"This result implied that simple bias-correction techniques of GCM data, which is addictive correction method to monthly mean runoff, may not be suitable for use in flood hazard estimation and that different bias-correction techniques should be tested (i.e. more focused on extreme values)."

*Figure 8: Can you specify the unit of measure of the x axis in (d) and (e)?*

**Reply: We thank referee #2 for comment. We will specify the x axis in (d) and (e).**

[Figure]

L488-503: I do not fully understand this analysis. Combining historical hazard maps with future flood frequency is basically the lookup method, right? (e.g. assuming that present-day 100-year RP will become 50-year RP in the future). Based on the outcome of the paper, I would rather conclude (here and in Conclusions) that historical hazard maps can be used as an indication of future hazard only if changes in flood frequency are properly accounted for.

**Reply:We thank referee #2 for comment.  As you pointed out, in general, estimating future flood frequency change and combining it with historical multiple-frequency flood hazard map is a good solution to construct the future hazard map, and it can be said as "lookup method". We added the below sentence in the end of the 1st paragraph of conclusion section:**

**"This implies that combining accurate historical hazard maps with information on future frequency changes of floods is considered optimal in general for generating future hazard map. Please note that the historical flood hazard map does not have to be r**

eanalysis-based simulation using GFM, and the proposed method can be also applicable to gauge-based or machine-learning based historical hazard map."

We'd like to note that the proposed look-up method applying extreme-value analysis at catchment-scale is beneficial at global scale studies since the frequency change varies from basin to basin and sub-basin to sub-basin.

---

## Author Response (AR1)

**In this document, the review comments are black color, our responses and our comment) are red color and the revised text in the revised manuscript are blue color.**

**\<Revised point other than review's comment\>**

Author's comment:

Please note that we have found errors in the exposure calculation procedures for reanalysis data results and Lookup method results in section 4.3 and have corrected them. Please also note that this does not affect the conclusions of this paper.

Specifically, we have reviewed Table2, Figure 9 and the inundation areas and affected population in the reanalysis data results and Lookup method in section 4.3.

Table 2: Inundation area of the 100-year RP hazard maps and affected population from the 100-year RP hazard maps constructed using various methods; "_no_modification" indicates a map constructed without Backwater_Modification. These results were obtained using IPSL-CM6A-LR.

| | | Inundation Area (Unit: Million km^2) | | Exposed Population (Unit: Million) | |
|---|---|---|---|---|---|
| | | all | depth > 5m | all | depth > 5m |
| **Historical Climate** | **Reference historical (Reanalysis)** | 18.38 | 2.19 | 1,625 | 216 |
| **Future Climate** | **GCM uncorrected** | 20.28 | 4.20 | 1,783 | 395 |
| | **Runoff Correction Method** | 20.45 | 4.02 | 1,802 | 406 |
| | **Lookup_Method_no_modification** | 19.49 | 3.23 | 1,821 | 417 |
| | **Lookup_Method** | 19.67 | 3.43 | 1,858 | 441 |

**\<Point-by-point response to the review #1\>**

Line 58: It would be good to add some references on in country hazard mapping. In general as this paragraph is an overview of local-scale flood hazard mapping efforts I found it a bit citation lite. It would be nice to signpost some other review articles as De Moel et al 2009 was published some time ago.

At the risk of self-promotion (no need to include any of this unless it's useful to support your later arguments) we included a UK focused review of local-scale modelling efforts that highlights some of the challenges of using these data sets in a climate conditioned application.

Bates, P. D., Savage, J., Wing, O., Quinn, N., Sampson, C., Neal, J., and Smith, A.: A climate-conditioned catastrophe risk model for UK flooding, EGUsphere [preprint], https://doi.org/10.5194/egusphere-2022-829, 2022.

Reply: We agree with referee #1 suggestion, so we have added "Bates et al., 2022" and "Wing et al 2022" as a reference.

Bates, P. D., Savage, J., Wing, O., Quinn, N., Sampson, C., Neal, J., and Smith, A.: A climate-conditioned catastrophe risk model for UK flooding, EGUsphere [preprint], https://doi.org/10.5194/egusphere-2022-829, 2022.

Wing, O. E., Lehman, W., Bates, P. D., Sampson, C. C., Quinn, N., Smith, A. M., Neal, J. C, Porter, J. R, and Kousky, C.: Inequitable patterns of US flood risk in the Anthropocene. Nature Climate Change, 12(2), 156-162, https://doi.org/10.1038/s41558-021-01265-6. 2022.

"In the research field, for example, Bates et al., 2022 estimated annual flood damage in UK and Wing et al., 2022 estimated flood damage in US by using local-scale flood-hazard maps." (Line 59-60 in revised manuscript)

"Line 81: assessment of future flood risks based on the spatial distribution of inundation depths has not yet been established" I agree at the global scale but there are many local scale examples of such studies that have had to consider the GCM bias issue. I think that needs to be acknowledged explicitly in this section or the previous paragraph.

Reply: We would like to express our gratitude to referee #1. We have revised the sentence as follows. We clarified that in this paper we described a "global" study. We also clarified that flood "risk" has been assessed, but flood "hazard" itself has not been sufficiently verified in previous global-scale studies.

"Although climatic and meteorological hazards under future climate change (e.g., extreme temperatures, droughts and heavy-rainfall events) have been widely assessed using direct output variables of general circulation models (GCMs), such as precipitation and temperature (Li et al., 2021, Lu et al., 2019), to date, no global high-resolution future flood hazard (i.e., inundation depth) has been sufficiently verified. Some studies have evaluated future flood risks (e.g., affected population and

GDP) at the global scale (e.g., Ward et al., 2020b); however, it is important to analyze global future flood hazards (i.e., inundation depth distribution), and to assess uncertainties such as those caused by various bias correction methods." (Line 80-86 in revised manuscript)

For example, this sentence "While bias-correction methods for precipitation and temperature have been studied in detail (Watanabe et al., 2012, Hempel et al., 2013 and Lafond et al., 2014), such methods have not been established for runoff data for use as inputs to flood models" is not correct without a more precise focus on the novelty of this study because there are many examples of localized climate conditions hydrological cascades all the way to inundation hazard and impact. These studies have some of the same issues you so I would have thought there were some useful conclusions from the literature that might feed into this section.

Reply: We have revised the sentence to clarify and focus on the novelty of this study. Specifically, we have changed from "such methods have not been established for runoff data for use as inputs to flood models" into the following sentence.

"such methods have not been established for runoff data for use as inputs to global flood models to construct large-domain future flood hazard maps." (Line 94-95 in revised manuscript)

Line 113: "Although CaMa-Flood is a global model, it is unique in that it represents the physical processes necessary to reproduce floodplain inundation dynamics" I appreciate there are unique aspects to CaMa-Flood, however there are several global flood inundation models so perhaps remove the word "unique" or be more specific about the combination of capabilities that are unique.

Reply: In response to referee #1 comments, we have removed "unique" and revise the sentence as follows.

"Although CaMa-Flood is a global model, it has characteristics that it represents the physical processes necessary to reproduce floodplain inundation dynamics" (Line 116-117 in revised manuscript)

Line 137: could you be specific about the simulation length – I suspect its shorter than most readers will assume by the wording long term. Is it 35 years?

Reply: As you understand, the simulation length is 35 years. Based on referee #1 comment, we have removed "long-term" and added "(time period: 1980-2014)" to the end of the sentence to specify about the simulation length.

"First, we conducted a historical river hydrodynamics simulation with a daily time step using observation-based runoff data (Reanalysis_Runoff) as an input to CaMa-Flood (time period: 1980-2014)." (Line 140-141 in revised manuscript)

Line 150: I have no issue with this pragmatic method for getting a spatially consistent hazard map, but I'm not convinced by the description. The hydrodynamic model simulated the backwater effect so I assume that this effect is captured in the hazard/level simulations. Could it be the case that the spatial inconstancy is therefore due to the distribution fitting and other data processing factors rather than a physical backwater effect being omitted? I appreciate the maps will look better with this correction but do you risk biasing the levels higher in the process? On a more fundamental level a hazard map is not an event, so does the water level even need to monotonically decrease downstream since you simple seek to best simulate the hazard rather than a plausible physical water surface which a hazard map will never be?

Reply: We thank referee #1 for comment. we have revised the sentence to make it clear that it is the case that the spatial inconstancy is due to the distribution fitting at each unit-catchment scale (grid-scale) as follows.

"If reverse-slope revision is not conducted, reverse slope produced by fitting the Gumbel distributions remains and the inundation-depth distribution may not be physically reasonable. For this reason, a novel reverse-slope revision method was applied in this study in the purpose of revising the spatial inconstancy caused by distribution fitting at each unit-catchment scale (grid-scale)." (Line 156-160 in revised manuscript)

In response to referee #1 comment "I appreciate the maps will look better with this correction but do you risk biasing the levels higher in the process?", we added the notes of overestimation to the last paragraph in section 4.2 (Specifically, the following sentence have been added).

"Please note that it is possible that it would be overestimated by applying Backwater_Modification because it performs corrections at all reverse slope occurrence points. Reversed water surface slopes can occur in conventional flood hazard maps, given that these maps are not always constructed by a single flood event simulation. Therefore, we conducted additional validation to investigate whether Backwater_Modification should be applied." (Line 457-460 in revised manuscript)

Section 2.3.1: Could you cite some alternatives to the bias correction method and explain why this one was chosen. I'm not an expert in these methods, but my assumption is that the results of this method would be sensitive to the choice of bias correction. Depending on your response above you might need to caveat the conclusions drawn around line 336.

Reply: We thank referee #1 for comment. We cited some alternatives to the bias correction method and explained why this one was chosen.

"Alternatives to the additive correction method include multiplicative correction method, which multiplies the ratio of GCM to the reanalysis data, and quantile-based mapping method (Panofsky and Brier 1968, Watanabe 2020), which obtains ordinal statistics from the reanalysis data and the GCM and creates an equation relating these statistics. In the future climate, the average monthly runoff may fluctuate significantly due to changes in the humid and arid zones and the timing of the wet and dry seasons caused by increasing temperatures. We selected the additive correction method because it is relatively insensitive to such fluctuations." (Line 197-202 in revised manuscript)

Line 383: "On the other hand, the lookup method did produce hazard maps consistent with the changes in flood risk under the future climate projected by the GCMs." I found this a bit confusing. Do you mean flood risk here? I assume you can only derive risk from the hazard maps, do you mean the changes in GCM runoff or precipitation? I think this section needs to be more precise about what you are looking for constancy with because I don't think its risk – or at least I didn't understand how it could be from the text.

Reply: As you pointed out, because we use "flood risk" incorrectly in the sentence, we have revised the sentence to clarify "the changes in flood hazard under the future climate projected by CaMa-Flood simulations". Specifically, we have changed from "the changes in flood risk under the future climate projected by the GCMs" into as follows.

"the changes in flood hazard under the future climate projected by CaMa-Flood simulations with GCM runoff input." (Line 394-395 in revised manuscript)

Line 390: "4.1 Which method is more convenient for generating future-hazard maps?" Convenient is the wrong word here it only applies to some of your arguments.

Reply: We agree with your comment. we have change from "convenient" into "reasonable" in the title of section 4.1.

"4.1 Which method is more reasonable for generating future-hazard maps?" (Line 401 in revised manuscript)

Line 394: I think you could also note here that this approach can also be used with hard mapping methods based on regional flood frequency analysis or machine learning from gauging station data, which at least in historically well monitored river reaches can be more accurate than rainfall-runoff reanalysis based methods (perhaps this is contested) for present day hazard mapping and extreme flow estimation because the modelling chain is much shorter (e.g. Laura Devitt et al 2021 Environ. Res. Lett. 16 064013 DOI 10.1088/1748-9326/abfac4)

Reply:

Thank you very much for your suggestion. As you pointed out, there is no need to limit it to reanalysis-basis as a "historical hazard map". Based on the above, we have added the following sentence to the end of 1st paragraph in section 5 "Conclusion".

Line 407 I think this is a separate point from the one above "The lookup method also has the advantage of facilitating
research on efficient construction of future climate hazard maps, as it allows for improvement of the reanalysis hazard map
by upgrading the model, and the estimated changes due to climate change can be considered separately" and you could
expand to discuss using multiple flood hazard models for the analysis as several are introduced in the preceding section.

Reply: We would like to express our gratitude to referee #1 suggestion. We added one point to advantage of the lookup
method.

"The lookup method also facilitates research on the efficient construction of future climate hazard maps because historical
hazard maps can be prepared separately from the estimation of future frequency change. This is beneficial for two aspects: 1)
it allows for improvement of the reanalysis hazard map by upgrading the model; 2) it allows for use of multiple reference
hazard maps by using different reanalysis-based simulations" (Line 420-424 in revised manuscript)

Section 4.2: I think it's valuable to discuss this correction (so I'm not suggesting a major revision). But in my opinion it's
debatable whether this correction is desirable for a hazard map which is not an event and thus not a physically possible water
surface. I think it potentially double counts for the backwater effect and over-predicts the flood inundation levels by biasing
locations upstream to any over-prediction errors in the distribution fitting downstream. That said the hazard mapping is
improved relative to some high quality validation data so I don't dispute that some form of postprocessing of the levels to aid
spatial constancy is the wrong thing to do… but you could smooth for example. Personally I would slightly modify the
discussion in section 4.3 to present the results with and without the backwater modification as an indicator of the sensitivity
to this issue – which is less than the climate change signal and impact of different GCM treatments. This would also require
a small edit to the conclusion ~line 532.

Reply: We would like to express our gratitude to referee #1. Backwater_Modificiation is applied in order to revise the spatial
inconsistency that is due to the distribution fitting (fitting Gumbel distribution) at the unit-catchment scale. In response to
referee #1 comment "I think it potentially double counts for the backwater effect and over-predicts the flood inundation levels by biasing locations upstream to any over-prediction errors in the distribution fitting downstream." we have added the notes of overestimation (Specifically, the following sentence will be added) in the last paragraph on section 4.2.

"Please note that it is possible that it would be overestimated by applying Backwater_Modification because it performs corrections at all reverse slope occurrence points. Reversed water surface slopes can occur in conventional flood hazard maps, given that these maps are not always constructed by a single flood event simulation. Therefore, we conducted additional validation to investigate whether Backwater_Modification should be applied." (Line 457-460 in revised manuscript)

Line 465: why choose lower resolution population data rather than say Worldpop? Could this bias the results?

Reply: We would like to express our gratitude to referee #1 question.

In this study, as to the estimates of future flood risk change, we compared the differences between the two methods to construct a future hazard map or with/without implementation of bias correction. Since the purpose is to analyze the impact of correction at the global scale, we think that 30 arcsec resolution was enough.

We used Gridded Population of the World; (CIESIN, 2018), which has data that are stored together for the entire world, rather than Worldpop, whose data are broken down by country.

In addition to above, we added the following notes regarding uncertainty in spatial resolution in section 4.3.

"As discussed by Zhou et al., 2021, the spatial resolution of a flood hazard map is a particularly important determinant of its value for impact assessment. Smith et al. (2019) evaluated the population exposure to a 1-in-100 year flood in 18 developing countries, and found that decreasing the spatial resolution of the flood hazard map from 90 to 900 m increased the exposure by 51-94 % for different population products. Although there is uncertainty involved in the choice of the spatial resolution of the flood hazard map, we selected 30-arcsec instead of 3-arcsec resolution to compare future flood hazard map construction methods on a global scale." (Line 497-502 in revised manuscript)

<Point-by-point response to the review #2>

L25: "...changes in flood risk..." I would correct in "changes in flood hazard"

Reply: As you pointed out, because we use flood risk incorrectly in the sentence, so we have revised it from "risk" to "hazard" as follows.

"On the other hand, the lookup method produced future-hazard maps that are consistent with flood hazard changes projected by CaMa-Flood simulations obtained using GCM runoff input, indicating the possibility of obtaining reasonable inundation-area distribution." (Line 24-26 in revised manuscript)

L29 "we discuss future changes at global scale..."

Reply: We have revised the sentence as follows.

"We discuss future changes at global scale in inundation areas and the affected population within the inundation area." (Line 29-30 in revised manuscript)

L49-50: "To elucidate the potential impacts of flood disasters, a high-resolution map of potential disaster impacts must be developed, commonly named a hazard map." This is not fully correct. In flood risk literature, hazard is a component of risk but it is not a synonym of disaster impact (see for instance Ward et al, 2020). Perhaps you could replace with "To elucidate the potential impacts of flood disasters, high-resolution maps of disaster impacts must be developed".

Reply: In order to make an appropriate sentence that leads to the commonly named a hazard map, we have revised the sentence as follows.

"To elucidate the potential impacts of flood disasters, high-resolution inundation-depth maps must be developed, commonly named a hazard map." (Line 50-51 in revised manuscript)

L70-72 "Uses of large-domain flood-hazard maps include estimation of the affected population within an inundation area and determination of the impacts of flooding on GDP and urban areas in the current climate". This sentence should be reworded. In literature, the maps including impacts such as population and/or urban areas exposed are generally called flood risk maps

Reply: As you pointed out, the sentence was not clear and was not stated correctly, so we have revised as follows to clarify the meaning of "Large-domain flood-hazard maps were used for many applications such as estimation of risks." Specifically, we have revised it as follows.

"Large-domain flood-hazard maps have been used in many applications such as estimation of the affected population within an inundation area and determination of the impacts of flooding on GDP and urban areas in the current climate (Ward et al., 2020a)." (Line 72-74 in revised manuscript)

L72-74: Bernhofen et al (2018) compared six global flood models against satellite-derived flood maps, so adding a reference here would be appropriate in my view.

Reply: We agree with your point. We have added Bernhofen et al (2018) to the sentence as follows.
"Hirabayashi et al., 2022, Trigg et al., 2016 and Bernhofen et al., 2018 compared multiple global flood models and analyzed the factors contributing to differences in inundation areas and depths." (Line 75-76 in revised manuscript)

L80-81 this sentence should be modified, because several flood risk assessments have been carried out at scales from global to local.
Reply: We have revised the sentence as follows. We clarified that in this paper we described a global study. Also, we
clarified that flood risk has been studied, but flood hazard itself has not been sufficiently verified.
"Although climatic and meteorological hazards under future climate change (e.g., extreme temperatures, droughts and heavy-rainfall events) have been widely assessed using direct output variables of general circulation models (GCMs), such as precipitation and temperature (Li et al., 2021, Lu et al., 2019), to date, no global high-resolution future flood hazard (i.e., inundation depth) has been sufficiently verified. Some studies have evaluated future flood risks (e.g., affected population and
GDP) at the global scale (e.g.,Ward et al., 2020b); however, it is important to analyze global future flood hazards (i.e., inundation depth distribution), and to assess uncertainties such as those caused by various bias correction methods." (Line 80-86 in revised manuscript)

L147-154: I think that the description of the post-processing method needs more detailing. In particular I have some questions:

- My understanding is that the authors fitted a Gumbel distribution on each pixel of the Camaflood 6-arcmin grid, correct? Or do you use different areas for the fitting?

Reply: We have revised the sentence as follows to clarify that fitting a Gumbel distribution on each pixel of the CaMa-Flood 6-arcmin grid.

"We fitted the Gumbel distribution (Zhou et al., 2021) to the time series of annual maximum river water levels using the L-305    moments method (Hosking, 2015) on each of the CaMa-Flood 6-arcmin grid." (Line 146-147 in revised manuscript)

- In lines 155-162 you state that water surface elevation is uniform within each 6-min unit catchment, so I assume that water level in upstream catchment are increased to the same water level of downstream catchment, right? If yes, please specify this in the text

Reply: Referee #2's understanding is correct. As referee #2 suggested that, we have revised the sentence as follows.

"To avoid this issue, if a reverse water slope was obtained in the water surface elevation distribution, we increased the water 315    surface elevation of upstream catchments to match those of downstream catchments." (Line 154-155 in revised manuscript)

- Can you also explain why this approach was not needed in previous studies based on CamaFlood?

Reply: We thank referee #2 for question. We would say the risk estimate of the previous studies could be improved by applying this reversal-slope modification approach. As described in the introduction, flood depth distribution itself had not been extensively evaluated in previous studies.

In addition, the purpose of applying reverse-slope revision is written more clearly as follows.

"If reverse-slope revision is not conducted, reverse slope produced by fitting the Gumbel distributions remains and the inundation-depth distribution may not be physically reasonable. For this reason, a novel reverse-slope revision method was applied in this study in the purpose of revising the spatial inconstancy caused by distribution fitting at each unit-catchment scale (grid-scale)." (Line 156-159 in revised manuscript)

L218: Alfieri et al (2017) actually employed the lookup method, because they used historical flood hazard maps coupled with changes in frequency under future climate scenarios.

Reply: We added Alfieri et al (2017) as a reference as follows.

"Alfieri et al., 2017 used the lookup method to estimate future affected population and damage by flood using historical flood hazard maps coupled with frequency changes under future climate scenarios." (Line 225-227 in revised manuscript)

L405-406: Based on the results, the authors could maybe draw the conclusion that standard bias-correction techniques of GCM data are not suitable for use in flood hazard estimation (having been developed for different types of climate studies), and that different bias-correction techniques should be used (i.e. more focused on extreme values)

Reply: We agree with referee #2 suggestion. In response to referee #2' suggestion, we revised the sentence in the section 4.1

as follows.

"This result suggests that simple bias-correction techniques of GCM data, i.e., additive correction to monthly mean runoff, may not be suitable for use in flood hazard estimation and that various other bias-correction techniques that focus on extreme values should be tested." (Line 418-420 in revised manuscript)

Figure 8: Can you specify the unit of measure of the x axis in (d) and (e)?

Reply: We thank referee #2 for comment. We specified the x axis in (d) and (e).

[Figure]

L488-503: I do not fully understand this analysis. Combining historical hazard maps with future flood frequency is basically the lookup method, right? (e.g. assuming that present-day 100-year RP will become 50-year RP in the future). Based on the outcome of the paper, I would rather conclude (here and in Conclusions) that historical hazard maps can be used as an indication of future hazard only if changes in flood frequency are properly accounted for.

Reply: We thank referee #2 for comment. As you pointed out, in general, estimating future flood frequency change and
combining it with historical multiple-frequency flood hazard map is a good solution to construct the future hazard map, and it can be said as "lookup method". We added the below sentence in the end of the 1st paragraph of conclusion section as follows.

"Thus, combining accurate historical hazard maps with information on future flood frequency changes may be optimal in general for generating future hazard maps." (Line 546-547 in revised manuscript)
We'd like to note that the proposed look-up method applying extreme-value analysis at catchment-scale is beneficial at global scale studies since the frequency change varies from basin to basin and sub-basin to sub-basin.